# Flood risk perception and adaptation capacity: a contribution to the socio-hydrology debate

Sven Fuchs[1], Konstantinos Karagiorgos[1], Kyriaki Kitikidou[2], Fotios Maris[3], Spyridon Paparrizos[4], Thomas Thaler[1]

[1] Institute of Mountain Risk Engineering, University of Natural Resources and Life Sciences, Vienna, Austria
[2] Department of Forestry and Management of the Environment and Natural Resources, Democritus University of Thrace, Orestiada, Greece
[3] Department of Civil Engineering, Democritus University of Thrace, Xanthi, Greece
[4] Faculty of Environment and Natural Resources, Albert-Ludwigs-University Freiburg, Germany

*Correspondence to*: Sven Fuchs (sven.fuchs@boku.ac.at)

**Abstract.** Dealing with flood hazard and risk requires approaches rooted both in natural and social sciences, which provided the nexus for the ongoing debate on socio-hydrology. Various combinations of non-structural and structural flood risk reduction options are available to communities. Focusing on flood risk and the information associated with it, developing risk management plans is required but often overlooking public perception of a threat. The perception of risk varies in many different ways, especially between the authorities and the affected public. It is because of this disconnection that many risk management plans concerning floods have failed in the past. This paper examines the private adaptation capacity and willingness with respect to flooding in two different catchments in Greece prone to multiple flood events during the last 20 years. Two studies (East Attica and Evros) were carried out, comprised of a survey questionnaire of 155 and 157 individuals, from a peri-urban (East Attica) and a rural (Evros) area, respectively, and they focused on those vulnerable to periodical (rural area) and flash floods (peri-urban area). Based on the comparisons drawn from these responses, and identifying key issues to be addressed when flood risk management plans are implemented, improvements are being recommended for the social dimension surrounding such implementation. As such, the paper contributes to the ongoing discussion on human-environment interaction in socio-hydrology.

**Keywords**: *flood risk management; adaptation capacity; risk awareness; risk perception; socio-hydrology*

## 1 Introduction

Increasing flood losses throughout Europe have led the European Commission to issue the 'Directive on the Assessment and Management of Flood Risks' (Commission of the European Communities, 2007) as one of the three components of the European Action Programme on Flood Risk Management (Commission of the European Communities, 2004). This directive requires the Member States to establish flood risk maps and flood risk management plans based on a nation-wide evaluation of exposure and vulnerability (Fuchs et al., 2017). While in recent years, considerable efforts have been made towards flood risk maps (Fuchs et al., 2009; Meyer et al., 2012), the requirements with respect to flood risk management and associated management plans are less-well studied (Mazzorana et al., 2012, 2013; Hartmann and Spit, 2016). Of particular importance seems the paradigm of public participation and societal adaptation in assessing local risks, and the legal and institutional settings necessary therefore (Hartmann and Driessen, 2013; Thaler and Levin-Keitel, 2016).

Insights into flood mitigation behaviour are essential because of the ongoing shift to risk-based flood management approaches, which require a contribution from flood-prone households to risk reduction (Bubeck et al., 2013). Generally speaking, risk perception influences the individual adaptation strategy through learning processes from past events (Bubeck et al., 2012; Collenteur et al., 2015). This so-called adaptation effect relates

to the development that frequent flood events may decrease individual vulnerability in the floodplain area
through the implementation of local structural protection measures (Holub et al., 2012; Jongman et al., 2014a; Di
Baldassarre et al., 2015; Mechler and Bouwer, 2015). The models proposed in the literature so far (see for
example Di Baldassarre et al., 2013a) focus mainly on catchment hydrology as well as associated long-term
response of human actions, such as incorporation of changes in demography, technology and society.
Nevertheless, short-term social aspects as one of the central points of societal adaptation are less well studied
(Keiler et al., 2005), but play a major role in social-hydrology with respect to an assessment of human-
environment interaction. The conceptual models, however, are so far relatively simplistic to mirror individual
responses and coping capacity (Temme et al., 2015). As such differences within a society, especially between
rural and urban areas as well as with respect to different flood types and frequencies still remain fragmentary.
Additionally, there is also evidence that sub-regional differences play an important role in the use of adaptation
strategies at household level (Higginbotham et al., 2014; Thaler and Priest, 2014; Thaler and Levin-Keitel,
2016). Acknowledging these findings, our paper explores differences in risk perception and individual response
to flood risk management strategies within two different sub-regional areas. Actions undertaken across urban and
rural farming populations characterised by different socio-economic conditions and affected by different flood
hazard types are studied, as well as their different response efficacy in flood risk management. This paper also
links management options assessed by individuals who belong to at-risk communities with direct experience of
floods in previous years, as well as the demographic profile of these individuals in terms of employment status,
education level, and gender. These variables – which focus on social behaviour and adaptation in flood risk
management – play a central role in the current socio-hydrology debate, but are so far repeatedly missed in the
literature (Gober and Wheater, 2015; Loucks, 2015). Therefore, a further step for including individual responses
and coping capacities in socio-hydrology models is made.

1.1 Coupled human-environment interaction in flood risk management

It is widely acknowledged that floodplains have always been attractive settlement areas, and, as a consequence,
people and assets are at risk of flooding. Dynamics behind the spatial and temporal pattern of exposure and
vulnerability are dependent on the spatial extent of flood hazards threatening societies, in particular their
magnitude and frequency, as well as on the socio-economic changes within society (Keiler et al., 2010). While
hazard assessment has a long tradition, the assessment of exposure and the quantification of vulnerability are
more recent concerns in hazard and risk research (Merz et al., 2010; Birkmann et al., 2013). Some aspects of
research in hydrology, such as the impact of highly destructive processes on buildings (Mazzorana et al., 2009;
Fuchs et al., 2012; Mazzorana et al., 2014), infrastructure (Zischg et al., 2005a, b) and agriculture (Morris and
Brewin, 2014), as well as challenges regarding multi-hazard risks (Kappes et al., 2012a, b) contribute to close
the gap between disciplinary approaches in science and humanities. Nevertheless, concepts of mitigation and
adaptation may remain fragmentary with respect to the optimal level of protection of exposed societies or
elements at risk (Ballesteros Cánovas et al., 2016). Moreover, most analysis has so far been based on a static
approach and neglect long-term as well as short-term dynamics in hazard, exposure and vulnerability (Fuchs et
al., 2013). Only recently such issues have been quantitatively analysed, such as shown by e.g. Jongman et al.
(2014b) for the Netherlands and Fuchs et al. (2017) for the European Alps.

Flood risk dynamics are linked to a trade-off 'between the memory of flooding events (which makes the
community move away from the river) versus the willingness to maximise economic benefit by moving close to
the river' (Di Baldassarre et al., 2013a, p. 3298). The context of dynamic flood risks is driving transformation
regarding the role of the state in responsibility sharing and individual responsibilities for risk management and
precaution (Mees et al., 2012; Adger et al., 2013). Emerging flood risk strategies place the lead responsibility on
local organisations to determine local strategies to manage local risks which demand societal transformation
(Driessen et al., 2013) in vulnerability reduction (Fuchs et al., 2011). The main reasons for this shift from
centralised to decentralised organisation is that local scale may be more efficient in dealing with risk and
emergency management. Societal transformation and social adaptation requires adaptive capacities and in-depth
knowledge on the perception of flood risk within communities. The perception of flood risk among different
parts of the population, i.e. citizens affected and inhabitants of flood plains, may differ and leads to different
levels of public participation in risk management strategies (Thaler and Hartmann, 2016; Thaler et al., 2016).

The main challenge for risk reduction is rooted in the inherently connected dynamic systems driven by both
geophysical and social forces, hence the call for an integrative management approach based on multi-disciplinary
concepts taking into account different theories, methods and conceptualisations (Fuchs and Keiler, 2013; Keiler
and Fuchs, 2016; Goudie, 2017). Strategies to prevent or to reduce losses from hydrological hazards have a long
tradition and started in the mediaeval times, however, concerted action was only taken in the outgoing 19[th]
century when official authorities responsible for flood protection were funded (Holub and Fuchs, 2009). A
century later, Burton et al. (1993) referred to continuously rising flood property losses during the 1970s and

1980s in the US and concluded that the development of floodplain management measures such as levees for flood protection and river training to increase discharge capacities was offset by the continued vulnerability of older buildings, roads and bridges. Already earlier, White (1936) discussed the limit of economic justification of flood protection, which has been confirmed by other studies such as Holub and Fuchs (2008) and Remo et al. (2012) showing that measures other than constructive flood protection may be more cost-efficient. There is a broad spectrum of flood risk management options, usually conceptualised as the flood risk management cycle consisting of mitigation, preparedness, response, and recovery (Carter, 1991; Merz et al., 2010). In particular mitigation and preparedness are targeted at reducing the (physical and social) vulnerability of exposed communities and to increase their resilience and coping capacity (Fuchs, 2008, 2009), in current debates addressed as socio-hydrology. The roots of such approaches trace back to very early influential works by the Chicago school (Kates, 1962; Burton and Kates, 1964; White, 1964). Spatiotemporal-based research into vulnerability to hydrological hazards began with attempts to explain the rising level of flood damage in the US in conjunction with unprecedented efforts and expenditures to control them (White, 1945; White et al., 1958). Some of White's most notable work (White, 1945) was a particular benchmark in stimulating subsequent studies, and involved the identification and classification of adjustment mechanisms for flooding, perceptions of natural hazards, and choice of natural hazard adjustments (Hinshaw, 2006). Hence, even before the leading work published by Starr (1969) geoscientists and engineers made an attempt to study human adjustments to risk and associated vulnerability. The main point in this early research was the differentiation between extreme natural events and regular flooding affecting communities, which provided material for the vulnerability discussion up to the present time (White et al., 2001). In particular non-structural adjustments, consisting of arrangements imposed by a governing body (local, regional, or national) to restrict the use of floodplains, or flexible adaptation to flood risk that do not involve substantial investment in flood controls, still remain central with respect to the contemporary management of hazards and vulnerability in many catchments. As such, there is still a need to understand the mutual relations between flooding and societal response as well as between the development within society and the resulting influence on floodplain dynamics (Di Baldassarre et al., 2013a; Viglione et al., 2014), which is largely linked to risk perception and studies on human-environment interction.

1.2 Linking flood risk, perception and adaptation

A low risk awareness of residents living in flood-prone areas is considered among the main causes of their low preparedness, which in turns generates inadequate response to the threat (White, 1973; Burton et al., 1993; Scolobig et al., 2012). Risk perception 'denotes the process of collecting, selecting and interpreting signals about uncertain impacts of events' (Wachinger et al., 2013, p. 1049), and is a very complex framework with multiple influencing factors (Fischhoff et al., 1978; Slovic 1987; Slovic, 2000; Plapp and Werner, 2006; Wagner, 2007). A general distinction is made between situational factors (such as individual experiences and socio-economic circumstances) and cognitive factors (such as personal and psychological components influencing individual behaviour in decision-making process). Therefore, risk perception provides individual interpretation of flood hazards and needs to be integrated in the formal decision-making process (Plattner et al., 2006; Barberi et al., 2008; Fuchs et al., 2009; Bradford et al., 2012). Many studies showed that personal experience is influenced by how exposed people recognise the likelihood of a hazard event, and the magnitude of those events, as well as their attitudes and beliefs concerning responsibilities for mitigation and loss compensation (Bubeck et al., 2012; Damm et al. 2013). In overall, risk perception and awareness demonstrate a central role in flood risk management discussion (Fischhoff, 1995; Renn, 1998; Slovic, 2000; Siegrist and Gutscher, 2006; Soane et al., 2010; Bradford et al., 2012; Bubeck et al., 2012, 2013; Wachinger et al., 2013; Pino González-Riancho et al., 2015; Kienzler et al., 2015; Babcicky and Seebauer, 2016). However, both terms are complex and controversially discussed, especially in terms of successful implementation of local structural protection measures (Karanci et al., 2005; Siegrist and Gutscher, 2008; Hall and Slothower, 2009; Jóhannesdóttir and Gísladóttir, 2010; Harries and Penning-Rowsell, 2011; Scolobig et al., 2012). The literature presents various myths and debates of both risk perception and awareness in flood risk management, especially the relationships between risk perception and awareness and the successful use of local structural protection measures and individual preparedness. Bradford et al. (2012), for example, demonstrated that the aspect of risk awareness shows no clear relationship with the individual preparedness in future flood events. Nevertheless, the authors found a clear relationship between flood experiences and preparedness. Similar results were also found by Harries and Penning-Rowsell (2011), Bubeck et al. (2013) and Kienzler et al. (2015), where people with flood experiences were more likely to undertake precautionary measures.

Nonetheless, experience of flood victims is only one aspect in the proactive action in flood risk management (Higginbotham et al., 2014). Whitmarsh (2008) argued that experiences have to be paired with the individual value and belief. Therefore, individual actions can also be associated with other factors, such as home ownership (Grothmann and Reusswig, 2006; Burningham et al., 2008), socio-economic status of individuals (Kreibich et al., 2011; Duží et al., 2015) or effective risk communication (Soane et al., 2010; Meyer et al., 2012; Bubeck et

al., 2013). On the other hand, on the individual side – social networks and knowledge (social capital), which communicate that the precautionary measures are useful or effective – demonstrate a much higher likelihood to undertake precautionary measures compared to past experiences (Lo, 2013; Poussin et al., 2014; Babcicky and Seebauer, 2016). Nevertheless, other scholars (such as Kellens et al., 2011 and Duží et al., 2015) demonstrated no significant relationship between one of these variables with the positive influence of individual preparedness. Furthermore, high risk perception will not necessarily lead to the successful implementation of local structural protection measures, as presented by different scholars (Karanci et al., 2005; Siegrist and Gutscher, 2006; Hall and Slothower, 2009; Jóhannesdóttir and Gísladóttir, 2010; Soane et al., 2010; Bubeck et al., 2013). In general, different explanations for this development are available, such as that people with experiences can underestimate the threat because they feel helpless during the event (Soane et al., 2010). Other reasons may be the financial burden, difficulty to understand and locate the hazard source as well as the difficulties to install local structural protection measures (Kreibich et al., 2011; Działek et al., 2013; Koerth et al., 2013; Kienzler et al., 2015), or lack of relationship between national authorities dealing with flood risk management and flood victims (Harries, 2013). In this line, a central aspect is the question of responsibility for flood risk management (Parker et al., 2007; Holub and Fuchs, 2009; Soane et al., 2010). In particular, the question about the implementation and payment of local structural protection measures seems to be crucial (Holub et al., 2012), as well as the overall concept used to reduce vulnerability and exposure (Fuchs, 2009; Fuchs et al., 2015).

## 2 Materials and methods

In this paper, we selected two different sub-regional areas in Greece characterised by two different types of flooding: low onset river flooding in the Evros catchment and rapid flash flood hazards in the East Attica region. Apart from these two different flood types, the selection of the study sites was made because of their contrasting socio-economic characteristics.

The river Evros is one of the largest in length of the Balkan peninsula. The total watershed area is 53,000 km$^2$ with 320 km river length and an average slope of 0.77%. About 66% of the total surface area is in the Bulgarian territory, about 28% in the Turkish territory and about 6% in the Greek territory. The Greek part of the river is a rural area of about 3,300 km$^2$ with a population of 85,000 concentrated in few small towns and villages. The river is known for a long series of serious and devastating flood events with high socio-economic costs and environmental impacts on the riparian communities and even on the national economies of the three neighbouring countries (Angelidis et al., 2010; Skias et al., 2013; see Fig. 1a). The area is dominantly rural oriented, where agricultural activities play a major role in the local economy. Besides the great importance of the river for the three riparian countries there are no common routes of collaboration between the states with respect to flood risk management. The complexity of the river is mainly due to political and historical reasons.

The second case study is the region of East Attica located east of Athens, which is characterised by flash flood events due to the prevailing climatic, geomorphologic, and anthropogenic conditions (Massari et al, 2014; Karagiorgos et al., 2016a, b; see Fig. 1b). The study area extends from the municipality of Oropos in the north to the municipality of Lavreotiki in the south and is subdivided into the provinces of Marathon, Mesogia and Lavriotiki. The district covers an area of 1,513 km$^2$ between sea level and 1,109 m a.s.l. with a plain hilly relief and a population amounting to 502,348 inhabitants (Hellenic Statistical Authority, 2011). The study area is characterised by extensive anthropogenic activities with settlements continuously growing for more than 30 years (Papathanasiou et al., 2012). The economic development of this area is closely related to the construction of the international airport of Athens in 2001. In the period 1998-2010, the annual rate of increase of building development was within a range of 5% to 30% (Sapountzaki et al., 2011). As reported by Mantelas (2010) the province of Mesogia has developed faster than any other area in Attica during the last 20 years. Specifically the urban land cover increased from 60 km$^2$ in 1994 to 75 km$^2$ in 2000, and to 125 km$^2$ in 2007. In other words, while the urbanised area had grown by 25% during 1994-2000, it grew by 66% during 2000-2007.

We conducted a questionnaire survey between June and November 2012, based on a door to door survey, with flood victims in two different sub-regions in Greece. In total we selected 312 interviewees, 155 respondents from the East Attica study area and 157 interviews from the Evros study area.

Based on a pilot study in East Attica (Karagiorgos et al., 2016b, c), the core of the survey was formed according to the following key questions: (1) socio-economic circumstances about the interviewee (such as gender, current job position, education, etc.), (2) social vulnerability (such as local embeddedness in the communities, social networks/social capital, household structure, etc.), (3) the impact and experience of the past flood events as well as about compensation, (4) risk constructions and awareness, and (5) responsibilities in flood risk management.

The questionnaires were distributed in the research areas by researchers trained for this survey. The distribution of the questionnaires was based on geographical criteria in order to represent the research areas. To provide a good spread of answers, pre-coded and prompted nature with a meaningful Likert-type scale were used. Data were analysed separately for the two research locations (rural and peri-urban area) using SPSS (Statistical Package for the Social Sciences) for Windows, version 21.0 (IBM SPSS Statistics 21 Documentation, 2015). Statistical significance tests were used through Mann-Whitney U test (Mann and Whitney, 1947), logistic regression (Cox, 1958) and Recursive Partitioning Analysis (Breiman et al., 1984) in analysing the differences about the perception of individuals in the peri-urban and the rural area as well as for impacts of several variables on risk awareness. Further, the tests were conducted in order to analyse the impacts of past flood events on the individual risk perception and awareness as well as the impact of past events on the likelihood to undertake precautionary measures.

**3 Results**

3.1 Demographic characteristics

Demographically, our sample profiles of Evros and East Attica were compared in Table 1. The selected sample was found to have a strong over-representation of males (75%), and older respondents (45%) for the Evros case study. Additionally, the high retirement rate for Evros (41%) reflects the age bias within the sample, while the unemployment rate is under-represented (1%) in compare to the population, which is also typical for the region with the result of a relative social homogeneity of the sample (similar to Steinführer and Kuhlicke, 2012). On the other hand, the East Attica sample fairly represents the population.

[insert table 1 about here]

3.2 Causation belief

We asked the interviews for the main roots of past flood events. Table 2 presents the results from the questioners, where a lack of structural measures being the most frequently listed reason for past flood events. Categorising the answers, 18.1% in Evros and 28.0% in East Attica identified the lack of protective constructions as one key factor for flood events. Additionally, in Evros, 18.1% saw the lack of maintenance of protective constructions as a central issue of ongoing flood events, while in East Attica, deforestation (61.8%), building in high-risk areas (55.4%), interventions on the riverbed (58.6%) respondents saw as central arguments for the past flood events. Therefore, most of the affected people listed anthropogenic factors as a central problem for past flood events; in contrast to the low onset flood events in Evros.

[insert table 2 about here]

3.3 Risk perception and awareness

Fig. 2 shows the results for evaluation of individual risk construction, distinguishing the sampling group into whether they were seriously affected in the past. One should expect that people who were evacuated should report perceiving the risk significantly higher than those who were not evacuated. In neither region, however, there was a significant difference between the evacuated and non-evacuated clusters with respect to risk perception (Mann-Whitney U tests: affected and non-affected people, p = 0.453 for Evros, p = 0.489 for East Attica). All the respondents in Evros and the majority in East Attica (53%) answered that they believe that a flood will happen again; from these respondents 69% in Evros and 63% in East Attica believe that a flood will happen in the next year, while 31% in Evros and 13% in East Attica believed that a flood will happen in the next two years. Risk communication processes embedded in local hazard knowledge (mainly from elderly people and flood experiences from neighbours and friends) and to a lesser extend also directly from the government through official training and information initiatives were the main reasons that respondents were aware of living in a dangerous area.

[insert fig. 2 about here]

Additionally, the Recursive Partitioning Analysis (Breiman, 1984), for the East Attica dataset showed that only the variable "income" has a significant impact on individual risk awareness; in fact, people with a higher income are more likely aware of the flood risk. Analysing the correlation between age and perception of the hydro-geological environment was found to be non-significant ($\tau = 0.063$ and $p = 0.355$ for Evros and $\tau = -0.019$, $p = 0.766$ for East Attica). In neither case, age demonstrate an increasing in risk perception.

3.4 Implementation of local structural protection measures

Table 3 and 4 presents the correlation matrixes for the different measured variables. A strong positive correlation can be found between the variables income and the use of local structural protection measures. In particular, the interviewees from East Attica responded positively between both variables ($r = 0.902$, $p < 0.01$). Also, the results from East Attica demonstrated a higher understanding of cause-and-effect relationships in comparison to the rural area of Evros, where the interviewees mainly blame the state for not having undertaken sufficient structural flood defence schemes. However, the Evros results showed that suffering material damages in the past, interestingly, did not correlate with any other variables.

[insert table 3 about here]

[insert table 4 about here]

In rural communities of Evros, where the sample had various experiences with periodical flooding, risk awareness was found to be significant positively correlated to the individual preparation (Kendall's tau correlation coefficient $\tau = 0.286$, $p = 0.000$). On the contrary, in the urban area of East Attica, the risk awareness was found to be uncorrelated to flood preparation ($\tau = -0.102$, $p = 0.120$). Nevertheless, the majority of respondents (72% and 67% for Evros and East Attica, respectively) stated that they feel safe against floods. In contrast, 25 % and 14% of the respondents, for Evros and East Attica respectively, consider their region being maximal at risk. However, only 24.8% of the sampling in Evros, but 73.4% of the respondents in East Attica undertook practical steps to protect their private property. Furthermore, in contrast to Harries (2013), fatalism play a much stronger role in the rural area of Evros compared to the semi-urban area of East Attica. In the latter case study, citizens were usually less likely involved in professions or skilled to response adequately and quickly to flood hazards, which typically can be found in rural areas. A key reason is the lack of relationship between a national authority dealing with flood risk management and flood victims with the outcome that flood victims take over the strategy of fatalism and blaming instead of increasing willingness to take precautionary measures (Harries, 2008, 2012). In particular, Tables 5 and 6 encourage this argument that in fact the public government has to lead the responsibility for the Greek flood risk management system. Main reasons for the low willingness are the low number of damages in the past (for East Attica see also Karagiorgos et al., 2016a, b), historical socio-economic developments (especially for the Evros region as a periphery border region with strong state support in the past 30 years) and the missing link between risk perception, previous flood experiences and preparedness (Bradford et al., 2012). On the other hand, and similar to other studies, such as De Marchi et al. (2007) or Steinführer and Kuhlicke (2007), the role of the citizens is marginal.

[insert table 5 about here]

[insert table 6 about here]

These results show the classical free rider problem, because citizens request a flood protection scheme without contributing to the actual costs, which raise the challenge and conflict of social justice and equity in flood risk management (Johnson et al., 2007; Thaler and Hartmann, 2016). Having been evacuated during a flood event had no differences in this statement (49% of evacuated and 50% of non-evacuated people in Evros thought strongly that the state should pay, and 75% of evacuated and 79% of non-evacuated people in East Attica thought strongly that the state should pay). The Mann-Whitney U test for the difference in ratings between evacuated and non-evacuated people gave $p = 1.000$, both for Evros and East Attica. These results were in straight line with the question of which flood risk management strategy should be followed. They also showed

that lay people indicated a strong tendency to hard flood defences, such as building new dikes and embankments, which were thought to be more effective than non-structural flood risk management concepts, such as an improvement of the local land use management plan or individually preparedness (see also Table 7). Also other studies, such as Felgentreff (2000, 2003) and Plapp (2004), found similar results where residents see structural defences as the most useful instrument in flood risk management. In Evros the key conflict issues are related to the unsolved transboundary cooperation in the region (more than 86.3%).

[insert table 7 about here]

**4 Discussion**

The increasing impact of human activities on hydrological dynamics has led to a growing interest in the study of socio-hydrology (Di Baldassarre et al., 2015). Focusing on such human-environment interaction, the findings within the presented study contributed to advance the understanding of risk management and preparedness in flood risk management, with a particular focus on two different types of hydrological hazards in a Mediterranean environment (Table 8). The variable personal experiences of flood incidents showed no influence in the willingness to take precautionary measures, which is different to the studies by Thieken et al. (2007), Kreibich et al. (2009, 2011), Bubeck et al. (2012, 2013) or Poussin et al. (2014, 2015). The rural sample showed a lower individual responsibility to undertake practical local structural protection measures in contrast to the semi-urban community, which is surprising because the communities in Evros were affected by several annually flood events in the past years. Therefore, also the adaptation effect could not be observed in the results since the observation that the occurrence of more frequent flooding is often associated with decreasing social vulnerability was not proven. This is in clear contrast to results provided by Bubeck et al. (2012) or Collenteur et al. (2015), especially for rural communities with large experiences on river floods.

Main reason is the individual perception and interpretation of risk. Kasperson et al. (1988) called this cognitive bias as a result of societal amplification of risk, whereabouts social structure and processes influence individual behaviour. Similarly, Wisner et al. (2004) reported that people who are economically and politically marginal are more likely to stop trusting their own methods for self-protection, and to lose confidence in their own local knowledge. In particular, the Evros respondents showed main concerns mainly against upstream conflicts with Bulgaria; instead of individual responsibility. This behaviour get intensify by the social institutions and organisations (Kasperson and Kasperson, 1996) in the Greek flood risk management policy. Consequently, the citizens of Evros were blaming the neighbourhood country instead of increasing their own resilience capacity at local level. Further, in contrast to Harries (2013), fatalism played a much stronger role in the rural area of Evros compared to the semi-urban area of East Attica, where usually citizens were less likely to be involved in professions or gained protected skills to response adequately and quickly to flood hazards; which we usually can find within the rural areas. A key reason is the lack of relationship between national authorities dealing with flood risk management and flood victims with the result that flood victims take over strategies of fatalism and blaming instead of increasing their willingness to take precautionary measures (Harries, 2012, 2013).

A central reason is the historical socio-economic development of the area as a periphery border region with strong state support in the past decades. In addition, the results showed that with respect to the perception of the hydrological environment, a surprising 32% for Evros and 39% for East Attica thought that their environment is not at all dangerous. Nevertheless, all the respondents in Evros and the majority in East Attica (53%) expressed their believe that flooding will happen again. On the other side, a correlation between age and perception of the hydrogeological environment was found to be insignificant; people did not seem to have more accurate perceptions for the environment they live in as they age. Many respondents did underestimate the hazard associated with flooding, both in the rural area with periodical flooding, and in the urban area with flash floods. Nevertheless, for many individuals within the study areas the recent events were still vivid within their memories, which has been described as availability heuristic (Tversky and Kahneman, 1973, 1974). Moreover, the sampling (especially for the rural areas) showed strong affect heuristic decision behaviour (Slovic et al., 2004). Therefore, action should be taken and appropriate methods should be developed by flood risk managers to best provide flood-related information in order to raise the appropriate awareness.

Based on our findings, there is an increased challenge in areas where communities believe that it is the flood risk agencies and emergency responders being solely responsible for the implementation of preventative measures, where the self-protection of individuals is far less important. Further, the East Attica sample saw new structural

protection measures as the key of flood risk management strategies instead of improving individual preparedness (White, 1945; Di Baldassarre et al., 2013b, 2015) – the non-occurrence of flooding did not lead to a substantial increase in social vulnerability and exposure to flooding. A larger emphasis was placed by residents upon measures to reduce the risk of flooding, rather than focusing on the improvement of better planning which could reduce settlement activities (such as construction of new buildings) in hazard-prone areas.

[insert table 8 about here]

**5 Conclusion**

Our results have shown that both, the levee effect as well as the adaptation effect have considerable different characteristics in the study sites. Besides, our results have shown that assumptions in socio-hydrology are highly complex, such as how different levels of memory influence risk awareness and how risk awareness is linked to adaptation response. Memory is accumulated via direct experience and is proportional to the actual damage experienced by individuals. However, flood experience alone is not sufficient to encourage local adaptation strategies, as shown in the Evros catchment.

Because of the different notion of risk between the general public and the scientific community, those who are responsible for developing and implementing flood risk management strategies need to understand and to include the individual risk construction of those affected people. It is due to a lack of understanding of the authorities in charge that flood risk management policies have failed in many places so far. This study represents a social approach and provides some explanations for this failure, and is targeted towards incorporating public perceptions in developing risk management plans. Although fear is often used to advocate an increase in risk perception, the results show that this is not a way to promote the desired response within the people; the majority feel safe against floods, while many people believe that their environment is not at all dangerous, both in the rural area with periodical flooding and the urban area with flash floods. Gathered through an innovative approach, the practical findings presented here will help to facilitate flood managers in their developments of national and local flood risk management strategies that integrate the complexity of individual risk perceptions, such as preparing risk communication strategies to raise awareness within the community. Whatever the emphasis in flood risk management is there is no doubt that its interest is not a study of the environment or of man per se (Kasperson and Kasperson, 1996; Turner II et al., 2003). It is argued that dealing with hydrological hazards and resulting adverse socioeconomic consequences requires methods and concepts rooted both in natural sciences (with respect to hazard assessment) and social sciences (with respect to exposure and vulnerability). As a corollary, there is a strong and transdisciplinary need towards studies of coupled human-environment interactions. The concept of socio-hydrology was introduced as 'a new science of people and water' (Sivapalan et al., 2012, p. 1270). The emerging field of socio-hydrology claims to explicitly focus on such interactions, above all to observe the co-evolutionary interaction between human development and hazard management (Sivapalan, et al. 2012; Di Baldassarre et al., 2013 a, b; Montanari et al., 2013), including various combinations of structural and non-structural flood risk reduction options available to communities (Holub et al., 2012; Loucks, 2015). Finally, the proposed methodological approach within the debate on socio-hydrology is to incorporate the individual response to different flood frequency (sudden vs. continuously), different socio-economic environment (semi-urban vs. rural) as well as type of processes (flash floods vs. river floods).

Flood risk management plans are becoming increasingly important for the European countries as these management strategies take in both the social factors and physical nature of risk, inherently calling for a coupled human-environment interaction approach. As such, if risk is quantified from a dynamic perspective and using approaches from coupled human-environment interaction, changes in the management strategies become obvious compared to traditional approaches of mitigation and adaptation. The coupled dynamics between hazards and exposure call for further studies in similar environments in order to test whether our results have to be interpreted in terms of singularities, and how the approach of socio-hydrology may be further used to enhance our understanding of underlying risk perception patterns. This allows to extend the current socio-hydrological concepts as well as to support practitioners in the development of enhanced flood risk management strategies at local level.

**Acknowledgements**
This work received funding from the Austrian Science Fund (FWF): P27400 as well as from the Austrian Climate and Energy Fund project SHARED (project number KR16AC0K13268). The authors highly appreciate the suggestions of M. Sivapalan, G. Di Baldassarre and another anonymous referee on an earlier draft of this manuscript.

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

Figure 1. Maximum monthly discharge (Evros river, Fig. 1a) and maximum daily discharge (Rafina torrent, Fig.
1b) for available time series. The event of 3-5 February 2011 was taken as a reference event within the Rafina
catchment, East Attica, since it was the event with the largest magnitude over the measurement interval (12.5
4 hours duration, 18 mm/hr. rainfall intensity, and 80 m^3/s and 56 m^3/s maximum daily discharge (04 and 05
February, respectively, see also Papathanasiou et al., 2013)). Please note that after the November-
December/2007 event the gauging station at Evros (Pythio) was destroyed and has not yet been reinstalled.
Therefore, the event of 16 November-02 December 2007 (3,400 m^3/s, an area of 52,800 km^2 affrected, 5
fatalities and around 300 people displaced, see Brakenridge 2016) was the reference for the case study of Evros,
where in general flooding occurs if discharge exceeds 2,500 m^3/s (Angelidis et al., 2010). Data source for
Rafina: Hydrological Observatory of Athens, http://hoa.ntua.gr/timeseries/d/897 (Rafina Fladar, access 04
October 2016); data source for Evros: Regional Authority of Eastern Macedonia and Thrace, see also Angelidis
et al., 2010.

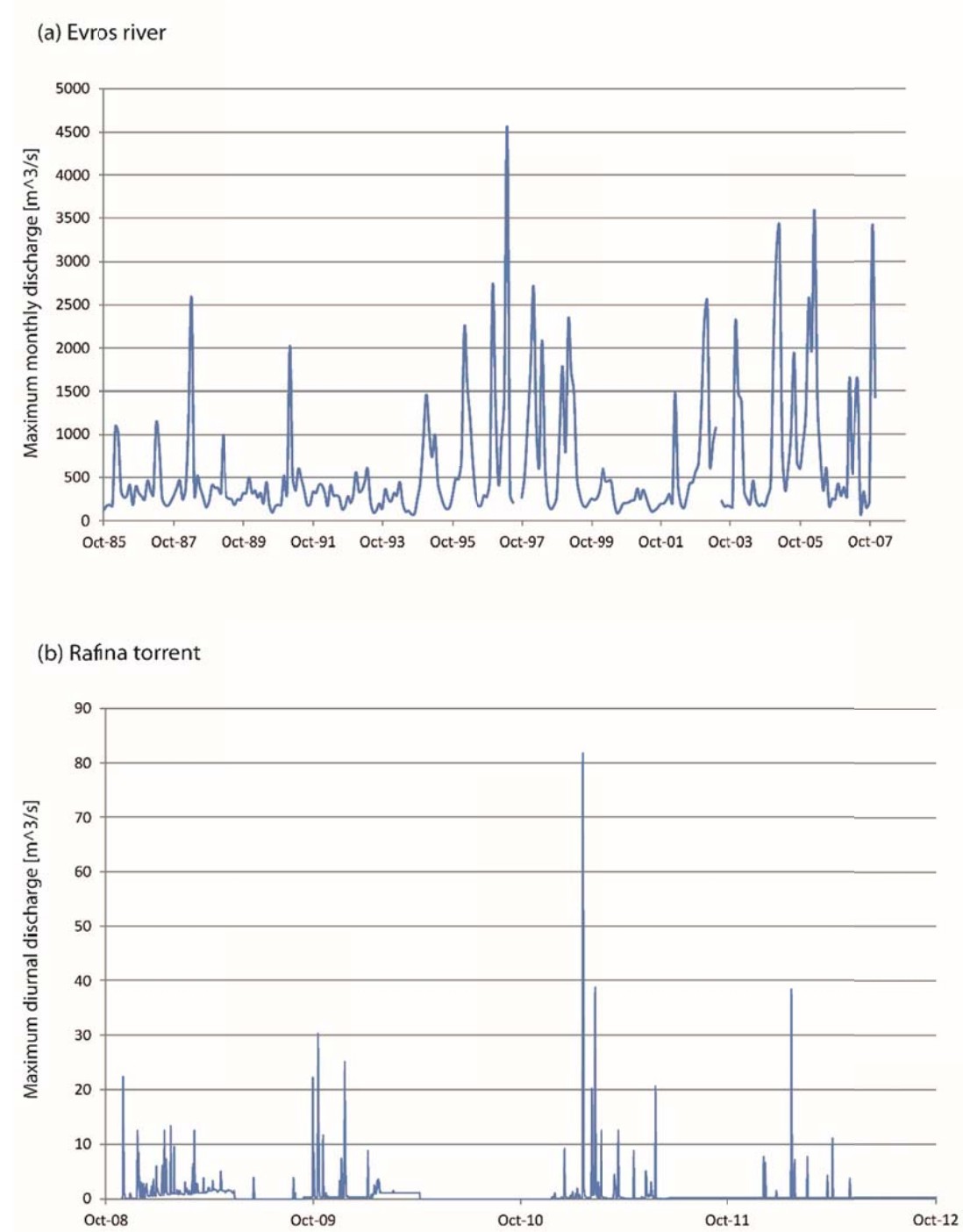

15

2    Figure 2. Local knowledge about hydro-geologically processes.

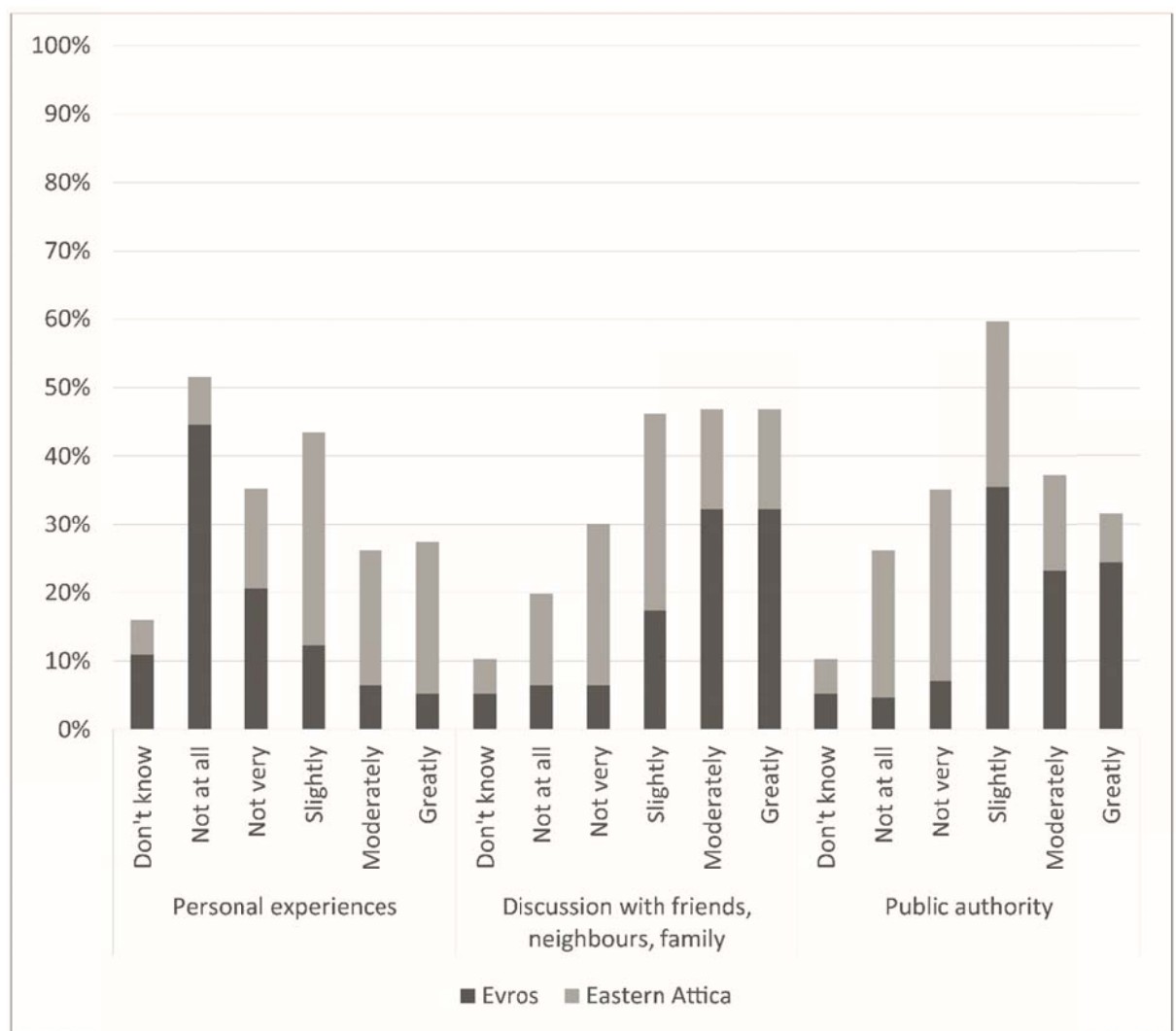

5

1    Table 1. Demographic characteristics in the study sites of East Attica and Evros

| Demographic variables | | East Attica | Evros |
|---|---|---|---|
| **Gender** | Male | 51.9% | 74.7% |
| | Female | 48.1% | 25.3% |
| **Education** | 1$^{st}$ level | 7.9% | 49.0% |
| | 2$^{nd}$ level | 57.9% | 45.0% |
| | 3$^{rd}$ level | 34.3% | 6.0% |
| **Employment** | Entrepreneur, free-lance, manager | 22.1% | 8.4% |
| | Trader, craftsman, farmer | 16.2% | 27.1% |
| | Teacher, employee, military | 29.9% | 7.1% |
| | Worker, store clerk, domestic collaborator | 10.4% | 6.5% |
| | Housewife | 1.9% | 5.8% |
| | Unemployed | 7.8% | 1.3% |
| | Retired | 3.9% | 40.7% |
| | Student or in search of first occupation | 7.8% | 0.0% |
| | Other | 0.0% | 3.2% |
| **Age** | <25 years | 5.1% | 2.0% |
| | 25-35 years | 24.8% | 4.7% |
| | 35-45 years | 24.2% | 6.7% |
| | 45-55 years | 23.6% | 14.0% |
| | 55-65 years | 15.3% | 28.0% |
| | ≥65 years | 7.0% | 44.7% |

1    Table 2. Respondents level of agreement as the causes of floods

| Activities | | East Attica | Evros |
|---|---|---|---|
| Deforestation | Don't know | 3.2% | 100.0% |
| | Not at all | 0.6% | 0.0% |
| | Not very | 3.2% | 0.0% |
| | Slightly | 12.7% | 0.0% |
| | Moderately | 18.5% | 0.0% |
| | Greatly | 61.8% | 0.0% |
| Building in risk areas | Don't know | 3.2% | 6.5% |
| | Not at all | 0.6% | 27.1% |
| | Not very | 5.7% | 10.3% |
| | Slightly | 17.2% | 10.3% |
| | Moderately | 17.8% | 17.4% |
| | Greatly | 55.4% | 28.4% |
| Lack of protective constructions | Don't know | 15.3% | 2.6% |
| | Not at all | 5.7% | 18.7% |
| | Not very | 21.0% | 12.9% |
| | Slightly | 16.6% | 27.1% |
| | Moderately | 14.6% | 20.6% |
| | Greatly | 26.8% | 18.1% |
| Lack of maintenance of protective constructions | Don't know | 14.6% | 5.2% |
| | Not at all | 8.3% | 17.4% |
| | Not very | 21.0% | 9.7% |
| | Slightly | 14.0% | 32.9% |
| | Moderately | 14.0% | 20.0% |
| | Greatly | 28.0% | 14.8% |
| Interventions on the riverbed | Don't know | 7.6% | 6.5% |
| | Not at all | 3.8% | 17.4% |
| | Not very | 5.7% | 7.7% |
| | Slightly | 5.7% | 30.3% |
| | Moderately | 18.5% | 23.9% |
| | Greatly | 58.6% | 14.2% |

Table 3. Correlation matrix East Attica

| | | 1 | 2 | 3 | 4 | 5 | 6 | 7 | 8 | 9 | 10 | 11 | 12 | 13 | 14 | 15 | 16 |
|---|---|---|---|---|---|---|---|---|---|---|---|---|---|---|---|---|---|
| 1 | Perception before last flood event | 1 | -.209* | -.293** | .102 | .091 | .182* | -.032 | .016 | .044 | .008 | -.047 | .180* | .232** | -.092 | -.058 | .122 |
| 2 | Evacuated at the event | | 1 | .199* | -.064 | -.176* | .023 | -.230** | -.248** | .032 | -.113 | -.057 | .025 | -.069 | -.069 | -.077 | -.079 |
| 3 | Suffered material damages | | | 1 | -.087 | .035 | .061 | .040 | .106 | -.120 | .088 | .250** | .193* | -.024 | .002 | -.028 | -.084 |
| 4 | Personal experiences | | | | 1 | .460** | .155 | -.305** | -.229** | .191* | .120 | .125 | .365** | .377** | .109 | .080 | .379** |
| 5 | Local knowledge | | | | | 1 | .245** | .210** | .264** | .165* | .180* | .220** | .030 | .087 | -.099 | -.099 | .091 |
| 6 | Official training and information initiatives | | | | | | 1 | .043 | .048 | -.036 | .056 | .132 | .146 | .189* | .163* | .099 | .127 |
| 7 | Personal precautions taken | | | | | | | 1 | .902** | -.265** | .184* | .323** | -.362** | -.396** | -.192* | -.161* | -.402** |
| 8 | Sufficient household income | | | | | | | | 1 | -.185* | .248** | .417** | -.332** | -.378** | -.211** | -.191* | -.363** |
| 9 | Period of living at the current residence | | | | | | | | | 1 | -.059 | -.203* | .010 | .148 | -.031 | -.010 | .110 |
| 10 | Retrospectively preparedness level | | | | | | | | | | 1 | .520** | .043 | -.031 | .034 | .058 | .066 |
| 11 | Present individual preparedenss level | | | | | | | | | | | 1 | -.061 | -.124 | -.063 | -.113 | -.157* |
| 12 | Deforestation causing the problem | | | | | | | | | | | | 1 | .652** | .400** | .350** | .504** |
| 13 | Construction of buildings in areas at risk causing the problem | | | | | | | | | | | | | 1 | .373** | .351** | .635** |
| 14 | Lack of structural devices causing the problem | | | | | | | | | | | | | | 1 | .917** | .502** |
| 15 | Lack of structural devices maintenance causing the problem | | | | | | | | | | | | | | | 1 | .509** |
| 16 | Interventions on rivers bed causing the problem | | | | | | | | | | | | | | | | 1 |

*. Correlation is significant at the 0.05 level (2-tailed).

**. Correlation is significant at the 0.01 level (2-tailed).

Table 4. Correlation matrix Evros

| | | 1 | 2 | 3 | 4 | 5 | 6 | 7 | 8 | 9 | 10 | 11 | 12 | 13 | 14 | 15 | 16 |
|---|---|---|---|---|---|---|---|---|---|---|---|---|---|---|---|---|---|
| 1 | Perception before last flood event | 1 | **-.507**$^{**}$ | **-.372**$^{**}$ | .001 | .006 | .118 | -.093 | -.061 | .125 | .009 | .060 | .$^{b}$ | **-.249**$^{**}$ | **-.204**$^{*}$ | **-.217**$^{**}$ | -.043 |
| 2 | Evacuated at the event | | 1 | **.363**$^{**}$ | -.074 | .065 | .115 | .013 | .125 | -.070 | .055 | -.004 | .$^{b}$ | **.265**$^{**}$ | **.209**$^{**}$ | **.183**$^{*}$ | .042 |
| 3 | Suffered material damages | | | 1 | -.116 | -.118 | -.095 | -.061 | .106 | -.146 | .030 | -.017 | .$^{b}$ | .150 | .043 | -.086 | -.147 |
| 4 | Personal experiences | | | | 1 | **-.286**$^{**}$ | **-.251**$^{**}$ | -.064 | -.051 | .132 | -.075 | -.121 | .$^{b}$ | **-.300**$^{**}$ | -.016 | .066 | .062 |
| 5 | Local knowledge | | | | | 1 | **.643**$^{**}$ | -.127 | -.058 | **.243**$^{**}$ | **.379**$^{**}$ | **.346**$^{**}$ | .$^{b}$ | **.242**$^{**}$ | .154 | .028 | -.129 |
| 6 | Official training and information initiatives | | | | | | 1 | -.058 | .101 | .103 | **.260**$^{**}$ | **.216**$^{**}$ | .$^{b}$ | **.328**$^{**}$ | .067 | **.168**$^{*}$ | .024 |
| 7 | Personal precautions taken | | | | | | | 1 | -.020 | -.050 | -.134 | **-.222**$^{**}$ | .$^{b}$ | .083 | .073 | **.196**$^{*}$ | **.194**$^{*}$ |
| 8 | Sufficient household income | | | | | | | | 1 | -.024 | .127 | .073 | .$^{b}$ | .103 | .060 | -.007 | -.103 |
| 9 | Period of living at the current residence | | | | | | | | | 1 | **.167**$^{*}$ | .135 | .$^{b}$ | .055 | -.031 | -.136 | -.101 |
| 10 | Retrospectively preparedness level | | | | | | | | | | 1 | **.523**$^{**}$ | .$^{b}$ | .091 | .125 | .020 | -.150 |
| 11 | Present individual preparedenss level | | | | | | | | | | | 1 | .$^{b}$ | .072 | .014 | .022 | -.071 |
| 12 | Deforestation causing the problem | | | | | | | | | | | | .$^{b}$ | .$^{b}$ | .$^{b}$ | .$^{b}$ | .$^{b}$ |
| 13 | Construction of buildings in areas at risk causing the problem | | | | | | | | | | | | | 1 | **.472**$^{**}$ | .153 | -.061 |
| 14 | Lack of structural devices causing the problem | | | | | | | | | | | | | | 1 | **.284**$^{**}$ | .113 |
| 15 | Lack of structural devices maintenance causing the problem | | | | | | | | | | | | | | | 1 | **.657**$^{**}$ |
| 16 | Interventions on rivers bed causing the problem | | | | | | | | | | | | | | | | 1 |

\*. Correlation is significant at the 0.05 level (2-tailed).

\*\*. Correlation is significant at the 0.01 level (2-tailed).

5   b. Cannot be computed because at least one of the variables is constant.

Table 5. Contributions to the costs for flood protection in East Attica

| | N | | *M* | SD |
|---|---|---|---|---|
| People at risk | 157 | 1=strongly disagree; 5= strongly agree | 2.401 | 1.386 |
| Local authority | 157 | 1=strongly disagree; 5= strongly agree | 3.815 | 1.363 |
| District | 157 | 1=strongly disagree; 5= strongly agree | 4.331 | 1.162 |
| Government | 157 | 1=strongly disagree; 5= strongly agree | 4.503 | 1.180 |

Table 6. Contributions to the costs for flood protection in Evros

| | N | Response scale | *M* | SD |
|---|---|---|---|---|
| People at risk | 155 | 1=strongly disagree; 5= strongly agree | 0.000 | 0.000 |
| Local authority | 155 | 1=strongly disagree; 5= strongly agree | 1.761 | 1.305 |
| District | 155 | 1=strongly disagree; 5= strongly agree | 3.226 | 1.506 |
| Government | 155 | 1=strongly disagree; 5= strongly agree | 3.955 | 1.369 |

Table 7. Perception of the effectiveness of adaptation measures

| Measures | East Attica | Evros |
|---|---|---|
| New protection works (such as levees or dams) | 79.6% | 2.0% |
| Ensure appropriate maintenance of existing protection works | 13.8% | 2.6% |
| Ensure better local land use management plans | 3.9% | 2.6% |
| Improve preparedness of people living in risk areas (e.g. information training drills etc.) | 2.6% | 6.5% |
| Other | 0.0% | 86.3% |

Table 8. Overview of the main results between both sub-regions

| | East Attica | Evros |
|---|---|---|
| Flood preparation | In East Attica (the urban area that experiences flash floods) risk awareness found to be uncorrelated to flood preparation. | In Evros (the rural area that experiences periodical flooding) risk awareness found to be positively correlated to flood preparation, i.e. the more aware, the more prepared. |
| Local structural protection measures | 73.4% of residents in East Attica made concrete steps to protect their family and property | A posteriori, 24.8% of residents in Evros made concrete steps to protect their family and property |
| Risk communication | The main reasons that respondents are aware that they are living in a dangerous area, where knowledge about hydro-geological phenomena is gained mainly by personal experience. | The main reasons that respondents are aware that they are living in a dangerous area, in Evros, are informal information, i.e. from family and friends, and formal information |
| Payments | 49% in East Attica believe that the state should pay for mitigation measures, while people who were evacuated and people who were not did not seem to be different. | A remarkable 77% in Evros believe that the state should pay for mitigation measures, while people who were evacuated and people who were not did not seem to be different. |