# Peer review of "Flood risk perception and adaptation capacity: a contribution to the socio-hydrology debate"

_Hydrology and Earth System Sciences, 2016_

## Referee Comment (RC1) · G. Di Baldassarre (Referee) · 24 Aug 2016

This paper deals with flood risk perception in Greece and discusses two contrasting case studies: a rural area exposed to river flooding in Evros, and a peri-urban area exposed to flash floods in East Attica. The topic of the paper is very relevant for the journal and the survey is overall scientifically sounded. However, I think that the presentation of the study and the discussion of the results should be improved. I report here my specific comments.

1) Introduction. The paper includes a nice review of the scientific literature in the field of flood risk awareness, perception and preparedness. Yet, this seems biased towards natural hazard research. To make the paper more relevant for the readers of this journal, I would suggest making a more explicit link to ongoing research in the hydrological community. The International Association of Hydrological Sciences, for instance, has recently launched a research decade called Panta Rhei, focusing on change in hydrology and society (Montanari et al., Hydrological Sciences Journal, 2013). Within this research initiative, there have been numerous efforts in exploring human-flood interactions and, more specifically, understanding if (and how and to what extent) flood experience is related to human response (Viglione et al., Journal of Hydrology, 2015; Di Baldassarre et al., Water Resources Research, 2015). I think it would be scientifically interesting to make a critical comparison of the (more empirical) outcomes of this Greek survey and the hypotheses made in these (more theoretical) studies.

2) Materials and methods. It is not entirely clear to me the timing and severity of past flood events in the two test sites. I think knowing them is crucial for a proper critical discussion of the results. How can I judge the impact of flood experience without knowing when exactly people in Evros and East Attica have experienced flood events? For example, it has been hypothesized that the impact of flood experience decays over time as the memory of floods tends to get lost (e.g. Di Baldassarre et al., Hydrology and Earth System Sciences, 2013). Thus, I suggest adding a new figure (or a table) with a time series of maximum annual flows, or peak over a threshold highlighting the timing of the survey. This would also make the paper more interesting for the hydrological community.

3) The Evros case study: Page 3, lines 25-35. Do we really need all these geo-political details? I am not sure, but I guess that some of them can be debatable. I would propose to shorten (or just delete) this part.

4) Causation belief. The text description of Table 2 ("categorising the answers...) does not seem to reflect what I see in the table. For instance, the greatest agreements about the causes of floods are reached for "building in risk areas" (28.4% in East Attica and 55.4% in Evros) and "deforestation (61.8% in Evros). Lack of protective constructions have a lower percentage in "greatly" agreement. I may be missing something here, but
I think the paper can be clearer on this point.

5) How was flood experience treated in the various correlation analysis if "there is a significant number of individuals without direct flood experience"?

6) Discussion. Very interesting concepts, such availability heuristic and affect heuristic, are mentioned in this section, but they are not sufficiently developed. My suggestion is to be more explicit and relate them to these specific case studies.

7) The bullet points summarizing the conclusions are too general and do not always reflect what was actually shown by this specific study. For example, number 2 states that "structural protection measures are an important measure to reduce flood risk". This is a very general statement, almost nobody would disagree with it. But, it is also vague - what do you mean by important? And, important to whom? - as well as, more concerning to me, unrelated to this specific case study. A better conclusion based on the survey presented here (Table 7) can be something like "citizens in East Attica and Evros perceive that new structural protection measures would be more effective than measures improving preparedness".

---

## Referee Comment (RC2) · Anonymous Referee #2 · 1 Sep 2016

To assess the flood risk perceptions and adaptation capacity is an important research topic which could help hydrologist, major readers of HESS, to understand how flood risk is perceived from the social perspective. The authors claim that "this paper links these self-assessed measures from individuals who belong to at-risk communities with direct experience with floods of previous years". This could be very useful to develop a social-hydrological solutions for flood adaptation. However, there are many ways in which this manuscript will be improved. Specifically,

1) the research aim is not straight forward, it is not very clear to me "what are your major research question?"

2) the questionnaires and this survey have to be improved. What are questions were

included in the questionnaires, why were they asked? how were they asked? how was the survey conducted? did you have a pilot survey? did you do sampling?.......These are central questions to be addressed.

3) how did the surveyed sample represent the population in terms of age and gender?

4) you did a comparative study between rural and peri-urban areas in Greece, some discussions on why they were different? and what are implications?

5) Figures and tables could be presented in a better way.

6) I can not draw your conclusions from your results.

7) What are implications of these survey results to hydrologists, who are major readers of this Journal?

---

## Author Comment (AC1) · 2 Sep 2016

Dear Professor Di Baldassarre,

thank you very much for your comments and suggestion to improve our paper. About your first idea to include this research into the current socio-hydrology or better say human-environment interaction debate is a great idea and we will include it as a theoretical framework and to apply it to our results.

Therefore, we will add two more chapters: first, introduction which deals with the debate within the socio-hydrology / human-environment interaction and a second one, which we would like to distinguish our conclusion and discussion chapter into one

chapter about discussion and one chapter about conclusion, whereabouts we would like to provide the feedback loop to our theoretical concept.

About the second comments: we will add some hydrological data to provide a better understand about, which type of flooding we will talk and how the different study sites were affected in the past years.

Third point: we short it, but it's a quite important issue for the results in the Evros study site. The political dimension lead the individual behaviour (response to past flood events) within this catchment and if we go back to the socio-hydrology debate (especially within your paper published in 2013 and 2015) the problematic of trans-boundary flood risk management and individual response to it, is not entirely covered, nevertheless, you covered the political dimension, but the points blaming, upstream-downstream or transboundary conflicts, implementation of EU Floods Directive be-tween EU and non-EU countries increase the complexity, which we try to answer in our paper. Therefore, the political dimension has to be included (but shorter) and we will open again the debate later within the discussion part.

Fourth point: true, we correct this part. We are very sorry for this confusion.

Fifth point: again, we correct this part, we only asked people who were affected from past events. Again, sorry for this mistake.

Sixth point: we largely re-wrote this part, first to show more the differences of how different types of floods (processes and characteristic) influence the communities and above all the response to it. Therefore, we split this chapter into discussion and con-clusion.

Last point: yes, we totally agree and we will largely re-wrote this part and also to provide a feedback loop to our new theoretical framework as well as to show the potential implication for policy makers and society also outside the two case studies.

---

## Author Comment (AC2) · 2 Sep 2016

Dear anonymous referee,

Thank you very much for your fast response and very fruitful comments and suggestions to improve our paper.

First point about the research questions: we will re-write the introduction section in order to clarify the gaps and to better show the research questions of our work.

Second point about the questionnaires, survey etc. This is very classical approach, used largely into the social science debate. The questions were asked with a door-to-door technique; about the sampling: we selected people based on to represent the

population within the two study areas as well as the recruiting is based on a snowballing technique. Nevertheless, of course we conducted a pilot survey with our students to test our questions. About the questions: we in total asked 76 questions, divided into following main sections: (1) socio-economic questions about the interviewee (such as gender, current job position, education etc.), (2) question on social vulnerability (such as local embeddeness in the communities, social networks/social capital, household structure etc.), (3) question on the impact and experience of the past flood events as well as about compensation, (4) risk constructions and awareness and (5) question on lessons learnt. We will clarify these concerns in the methods section.

Third point: the selection of the sampling fits within the socio-economic structure within the two case studies, especially in terms of gender and age. We will clarify this in the methods section.

Fourth point: this indeed is a weak point, which we will solve in the next version of the manuscript once we have the general acceptance of the responsible Editor to do so.

Fifth point: since the current figures and tables are in the line within the social science domain and therefore are state-of-the-art, we honestly see only little chances for a further improvement.

Sixth point: once we have the general acceptance of the responsible Editor, we will conduct a new chapter with the title conclusion and provide a more fruitful discussion how our study is linked and can contribute to the current socio-hydrology debate as well as to show the policy implication outside our study sites.

Last point: This paper provides a further step within the socio-hydrology discussion, which is currently more a theoretical concept with the first tries to translate it with empirical research. Firstly, we use the socio-hydrology debate to analyse and to assess our empirical data and provide a next step within the theoretical discussion; therefore, this paper provide will also provide the link to the Panta Rhei discussion. Further, the paper focus on following aims and scope of the journal: water-related natural hazards

and the interaction between hydrological and societal processes within the earth system

---

## Author Response (AR1)

Dear Editor,

Please find attached our revised version of the manuscript, as well as the answer to the referees.

Dear Professor Di Baldassarre,

thank you very much for your comments and suggestion to improve our paper. To oppose this research to the current socio-hydrology debate (which equals over large parts the human-environment interaction debate in geosciences) is indeed promising. Therefore we have re-written major parts of our contribution (Introduction and Discussion/Conclusion), and we also changed the title to better mirror the content of our manuscript. We strongly believe that our contribution will now better fit (a) to the overall discussion in socio-hydrology as well as (b) to the requested discussion on practical implications for hydrologists.

Second, we understand your concern about the timing and severity of past flood events in the two test sites. Therefore, we added a new Figure 1 showing the maximum monthly (Evros) and daily (Rafina catchment in East Attica) discharge in order to provide the potential reader with more information about the flood characteristics. Unfortunately, during the 2007 events in Evros the gauging station was destroyed and has not yet been fully recovered which means that the time series ends in 2007 (compare Figure 1). Nevertheless it becomes clear from the data how different the flood characteristics are, and that in both of the catchments a certain experience with frequent events is present.

Third, we shortened the "geopolitical issues" from the Evros case study description since we agree that this information may not be useful in the context of our study. The transboundary management question is discussed in the final part of our manuscript, but with more emphasis on the socio-hydrology debate.

Fourth, we changed the wording with respect to the description of Table 2 so that the content will become more accessible. Please be aware that in the previous version prepared for HESSD there were some minor mistakes in some of the Tables which were corrected in the new version. Therefore, the string of argumentations slightly changed for some of the studied parameters.

Fifth and sixth, as we changed the discussion section in order to mirror the comments on socio-hydrological research, the statements were corrected. Moreover, your fifth concern was due to a misunderstanding (or mre precise, a mis-formulation) since we only asked people who were affected from past events.

Finally, as we changed the conclusion section we do not have any bullet points (which indeed were shortening our messages in a way that was hard to understand). The final section is now connected to the introduction in order to show the potential implication for policy makers and society also outside the two case studies.

Dear anonymous referee,

Thank you very much for your fast response and very fruitful comments and suggestions to improve our paper.

Firstly, as we entirely re-wrote the manuscript with a stronger focus on the debate in socio-hydrology, we also sharpened the research question. Please see page 4, lines 1-17: "Taking these findings as basis for discussion, the present paper explores differences in risk perception and individually response to flood risk management within two different sub-regional areas. Different actions undertaken across urban and rural farming populations characterised by different socio-economic conditions and affected by different flood hazard types are studied, as well as their different response efficacy in flood risk management. This paper also links management options assessed by individuals who belong to at-risk communities with direct experience with floods of previous years, as well as the profile demographic of the individuals in terms of employment status, education level, and gender. These variables – which concentrate on the social behaviour in the flood risk management discussion – play a central role in the current socio-hydrology debate, but are so far repeatedly missed in the literature (Gober and Wheater, 2015; Loucks, 2015). The models proposed in the literature so far (see for example Di Baldassarre et al., 2013b) focus mainly on catchment hydrology as well as associated long-term response of human actions, such as incorporation of changes in demography, technology and society. Nevertheless, social aspects as one of the central points within the assessment of human-environment interaction play a major role in social-hydrology. The conceptual models, however, are so far relatively simplistic to mirror individual responses and coping capacity. As such differences within a society, especially between rural and urban areas as well as with respect to different flood types and frequencies still remain a gap. Therefore, in this paper a next step for incorporating individual responses and coping capacity into socio-hydrology models is proposed."

Secondly, you had some concerns about the questionnaires and the survey which are explained in detail in section 2 (materials and methods); please compare page 4, lines 21 ff.

Thirdly, the selection of the sampling fits within the socio-economic structure within the two case studies, especially in terms of gender and age. We also discussed this in the results section (please see page 5, lines 13 ff.).

Fourthly, we added some more information on the differences between rural and peri-urban areas; please see also the results section (page 6, lines 5 ff.) as well as the new discussion section (page 7, lines 11 ff.).

Fifthly, since the current figures and tables are in the line within the social science domain and therefore they are state-of-the-art, we honestly did not see any chances for a further improvement.

Sixthly, as we have re-written major parts of our contribution (Introduction and Discussion/Conclusion), and we also changed the title to better mirror the content of our manuscript, we strongly believe that our contribution will now better fit (a) to the overall discussion in socio-hydrology as well as (b) to the requested discussion on practical implications for hydrologists (see also your last point in the response to the discussion paper).

[revised manuscript text omitted]

---

## Author Response (AR2)

Editor Decision: Publish subject to minor revisions (further review by Editor) (10 Apr 2017) by Fuqiang Tian

Dear Authors,

I realised that this paper underwent a long reviewing process. One reason is the cross-discipline nature of the research, and another reason is, however, the diffuse depiction. I also realised that the core content in this paper is rather useful for the sociohydrological study, it provides a vivid case from the perspective of social science, and the results are of significance for the flood management. But the authors need to submit a thoroughly revised version by following all the comments and I will make a further editor review.

Best,

Fuqiang

Di Baldassare:

"Accept as it is"

My concerns about the paper were addressed and the revised paper is much more appealing to HESS readers. So, it can now be accepted for publication.

Anonymous #2:

"Reject"

Regrettably, my recommendation is to reject this manuscript to be published in HESS. The reasons are:

1) The authors added very extensive literature review on socio-hydrology, but the literature review is not internally well linked and some parts are not directly related to the methods and results sections.

*Response: We re-organised the manuscript to better link the Introduction to the material presented.*

2) The research aim is still not clear to me.

*Response: We re-organised the manuscript to better link the Introduction to the research aim.*

3) The authors added some details on how they implemented their studies, but the most important, why they designed their questionnaire and why through this questionnaire the research questions could be well answered, are totally missed.

*Response: The research on risk perception, social vulnerability and adaptation capacity is largely supported by quantitative questionnaires within the literature; see for example work conducted by*

*Anna Scolobig, Philipp Bubeck, Christian Kuhlicke or Thorston Grothmann and many more. The selection of questions within these questionnaires focussed on the assessment of the possibly drive mitigation behaviour and applied variables of psychological concepts to explain decision making to response to threats.*

4) The hydrological implication from the findings of this study is not well stated.

*Response: We have re-organised the manuscript in order to better show the take-home messages with respect to the research field of socio-hydrology.*

5) As I said in last version, there are many ways to improve the quality of result presentation. Unfortunately, the authors have not made any changes.

Sivapalan:

"Minor revisions"

I was not one of the original reviewers of the previous version of the article. Overall this paper makes a good contribution to socio-hydrology and explores the not of dynamic flood risk. This is highly topical, and I am positively disposed towards acceptance of the paper. However, I found the language somewhat diffuse, the storyline not well developed, and the take home message not as clear as I would have liked. Language may have contributed to it, but also the absence of a theoretical framework also contributed to the difficulty following the paper logically. Improved structure, with carefully chosen titles and subtitles and key sentences in key places to track the progression of ideas would help.

So if there is an opportunity, and I do not insist on this, I recommend that the authors undertake minor revisions to make the presentation better. This can be extremely helpful to raise the impact of the paper. In these revisions the authors should focus on the storyline and must make the take home message unmistakable.

In general I really enjoyed reading the paper and congratulate the authors for the work they have done, and thank them for their contributions to socio-hydrology.

*Response: The authors would like to thank for this very positive response. We change the structure to increase the reader friendliness within our paper and to avoid misunderstandings. Further, we added more sub-titles to help the reader following the structure.*

[revised manuscript text omitted]